# Synergistic interactions with PI3K inhibition that induce apoptosis

Yaara Zwang[1,2], Oliver Jonas[3†], Casandra Chen[1,2†], Mikael L Rinne[1,2], John G Doench[1], Federica Piccioni[1], Li Tan[4,5], Hai-Tsang Huang[4,5], Jinhua Wang[4,5], Young Jin Ham[4,5], Joyce O'Connell[1,2], Patrick Bhola[2], Mihir Doshi[1,2], Matthew Whitman[3], Michael Cima[3,6], Anthony Letai[2], David E Root[1], Robert S Langer[3,7], Nathanael Gray[4,5], William C Hahn[1,2,8*]

[1]Broad Institute of Massachusetts Institute of Technology and Harvard, Cambridge, United States; [2]Department of Medical Oncology, Dana-Farber Cancer Institute, Boston, United States; [3]The David H. Koch Institute for Integrative Cancer Research, Massachusetts Institute of Technology, Cambridge, United States; [4]Department of Cancer Biology, Dana Farber Cancer Institute, Boston, United States; [5]Department of Biological Chemistry and Molecular Pharmacology, Harvard Medical School, Boston, United States; [6]Department of Materials Science, Massachusetts Institute of Technology, Cambridge, United States; [7]Department of Chemical Engineering, Massachusetts Institute of Technology, Cambridge, United States; [8]Department of Medicine, Brigham and Women's Hospital, Harvard Medical School, Boston, United States

*For correspondence:
william_hahn@dfci.harvard.edu

†These authors contributed equally to this work

**Abstract** Activating mutations involving the PI3K pathway occur frequently in human cancers. However, PI3K inhibitors primarily induce cell cycle arrest, leaving a significant reservoir of tumor cells that may acquire or exhibit resistance. We searched for genes that are required for the survival of PI3K mutant cancer cells in the presence of PI3K inhibition by conducting a genome scale shRNA-based apoptosis screen in a *PIK3CA* mutant human breast cancer cell. We identified 5 genes (*PIM2, ZAK, TACC1, ZFR, ZNF565*) whose suppression induced cell death upon PI3K inhibition. We showed that small molecule inhibitors of the PIM2 and ZAK kinases synergize with PI3K inhibition. In addition, using a microscale implementable device to deliver either siRNAs or small molecule inhibitors in vivo, we showed that suppressing these 5 genes with PI3K inhibition induced tumor regression. These observations identify targets whose inhibition synergizes with PI3K inhibitors and nominate potential combination therapies involving PI3K inhibition.

## Introduction

The phosphatidylinositol 3-kinase (PI3K) pathway is frequently activated in breast cancers due to (i) amplification of *ERBB2*, an oncogene that stimulates the PI3K pathway and (ii) activating mutations of the PI3K catalytic subunit, *PIK3CA* (*Koboldt et al., 2012*). In addition, other genetic aberrations can lead to the activation of the PI3K pathway including *PTEN* deletion or loss-of-function mutations, *PIK3CA* amplification and activating *AKT* mutations. Constitutive PI3K pathway activation promotes cell proliferation and survival, and previous reports have demonstrated that tumors harboring mutations that activate the PI3K pathway require constitutive signaling of this pathway for tumor maintenance. Specifically, tumors that harbor mutant *PIK3CA* alleles exhibit significant dependence on *PIK3CA* expression and activity (*Cheung et al., 2011; Liu et al., 2011; Samuels et al., 2005*). In addition, oncogenic activation of *PIK3CA* leads to intrinsic resistance of HER2-positive breast cancer

**eLife digest** When cells become cancerous, they accumulate mutations in their DNA that switch on some genes at the wrong time and to higher levels than normal. These over-active genes help cancer cells to survive, grow and evade death. One of the genes that is often mutated and over-active in breast cancer encodes an enzyme called PIK3CA.

There are several drugs that bind to and inhibit the mutant version of PIK3CA. Recent experiments show that inhibiting over-active forms of this enzyme can stop cancer cells from growing, but it does not cause them to die. This means that the cells have the opportunity to become resistant to the drug, which can subsequently lead to tumor relapse. Therefore, researchers have been looking for other drugs that, when combined with the PIK3CA-inhibiting drug, are able to kill the cancer cells.

The first step to developing such a therapy is to identify genes that are essential for cancer cells to survive when they are exposed to the PIK3CA-inhibiting drug. One way to achieve this is to test what happens when you switch off individual genes one by one in these cells, an approach known as a functional genomics screen. Zwang et al. used this approach to identify genes in human breast cancer cells that have the potential to be useful drug targets. The screen identified five genes that can be individually switched off to kill the cancer cells.

Two of these five genes encode enzymes known as PIM2 and ZAK. Zwang et al. went on to find drugs that inhibit PIM2 and ZAK. As expected, administering these drugs together with the PIK3CA inhibitor caused breast cancer cells to die. Further experiments will be necessary to find out what roles these five genes play in breast cancer cells. In the future, these findings may lead to the development of more effective therapies for human cancers in which PIK3CA is over-active.

cells to HER2 inhibition (*Berns et al., 2007*; *Hanker et al., 2013*), and is more frequently activated in patients that exhibit acquired resistance to HER2 inhibition (*Chandarlapaty et al., 2012*).

The prevalence of PI3K pathway activation in breast cancer and its importance to cancer cell proliferation and tumor survival make targeting this pathway an attractive therapeutic approach. However, inhibition of the PI3K pathway often leads to proliferative arrest rather than cell death (*Elkabets et al., 2013*; *Klempner et al., 2013*; *Serra et al., 2008*) and to date has shown limited clinical benefit. Specifically, PI3K/AKT/mTOR inhibitor therapy induced a partial response in 18–30% of patients whose tumors harbor *PIK3CA* and/or *PTEN* mutations (*Janku et al., 2014*, *2013*, *2012*). Although this rate of partial responses was significantly higher than that achieved following treatment with therapies other than PI3K/AKT/mTOR inhibitors, this response was not associated with an improvement in either progression-free or overall survival of treated patients. Combination therapy consisting of Trastuzumab and Buparlisib, a PI3K inhibitor, resulted in a 17% partial response (*Saura et al., 2014*), and mTOR inhibition combined with aromatase inhibitors in patients with hormone-receptor positive advanced breast cancer showed extended progression-free survival (*Baselga et al., 2012*). Together, these studies suggest that targeting the PI3K pathway alone is only partially effective clinically.

We hypothesized that identifying targets whose inhibition in the context of PI3K inhibition leads to cell death would provide a foundation to develop combination therapies. Here using a genome-scale loss of function screen, we identified genes whose suppression induces cell death only in the presence of PI3K inhibition both in vitro and in vivo.

## Results

### A genome scale shRNA screen identifies genes whose suppression facilitates cell death in the setting of PI3K inhibition

To identify genes whose suppression converts the cytostatic response to PI3K inhibition into a cytotoxic response, we performed a positive-selection genome scale shRNA screen (*Figure 1A*) using MDA-MB-453 breast cancer cells, which harbor a *PIK3CA* H1047R mutation and *ERBB2* amplification. Treatment with the PI3K inhibitor GDC0941 leads to a complete proliferation arrest (*Figure 1—*

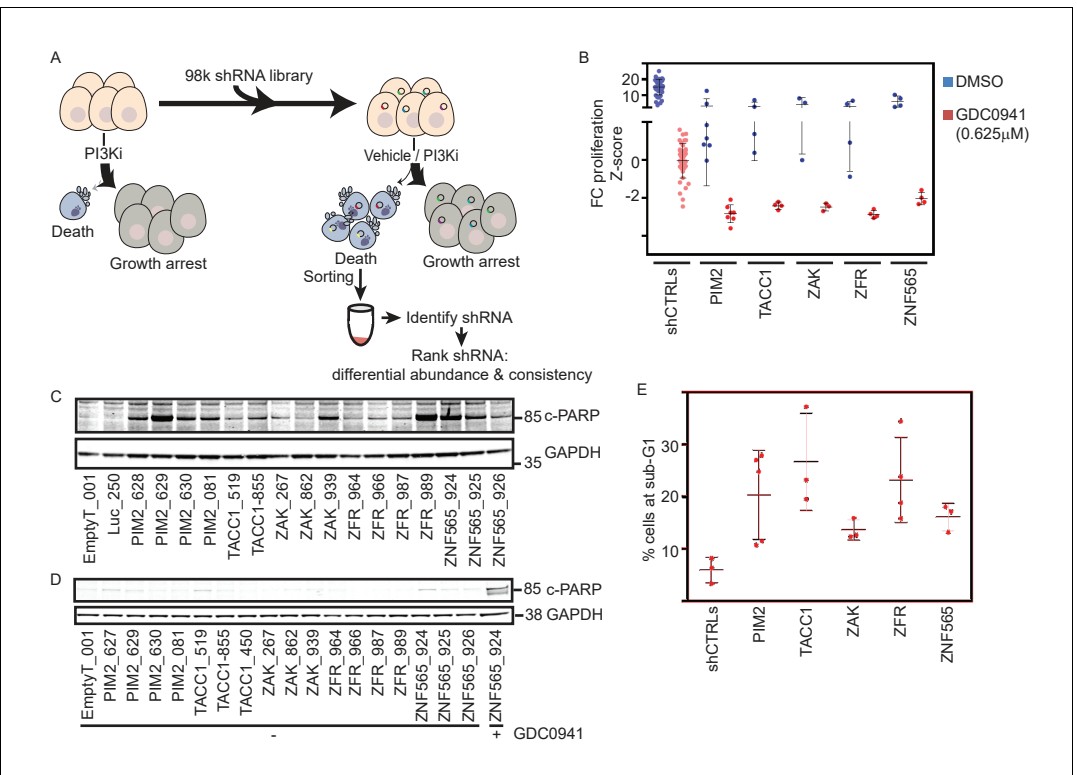

**Figure 1.** Genome scale shRNA screen identifies genes whose suppression facilitates PI3Ki-induced cell death. (A) A schematic representation of the pooled shRNA screen design. (B) Z-scores for fold-change of proliferation of MDA-MB-453-eGFP cells infected with multiple shRNAs targeting the indicated genes and treated for 9 days with GDC0941 (0.625 µM; red), or vehicle (DMSO; blue). Cells infected with five different control shRNAs (shCTRLs) were used to calculate Z-scores. Bars indicate standard deviation between the different shRNAs targeting each gene. Data shown are representative of three independent experiments. (C–D) MDA-MB-453 cells were infected with the indicated shRNAs, and then treated for 4 days with GDC0941 (0.625 µM) (C) or left untreated (D). Adherent and floating cells were collected and subjected to immunoblot analysis for induction of PARP cleavage. Cells infected with a shRNA targeting *ZNF565* and treated with GDC0941 (0.625 µM) for 4 days were used as positive control for PARP cleavage (D). Data shown are representative of two independent experiments. (E) MDA-MB-453 cells were infected as in B and treated for 4 days with GDC0941 (0.625 µM). Adherent and floating cells were collected and analyzed for DNA content by flow cytometry. Bars indicate standard deviation between the different shRNAs targeting each gene.

The following figure supplement is available for figure 1:

**Figure supplement 1.** Supporting information for shRNA screen setup and scoring.

*figure supplement 1A*) and suppression of AKT activity (*Figure 1—figure supplement 1B*) with minimal basal- and PI3Ki-induced cell death (*Figure 1—figure supplement 1C–D*).

We introduced a pooled lentivirally-delivered shRNA library in quadruplicate into MDA-MB-453 breast cancer cells. Cells that incorporated shRNAs were selected by resistance to puromycin and propagated in culture to deplete cells containing shRNAs targeting essential genes. Seven days post-infection, cells were treated with either the pan-PI3K inhibitor GDC0941 (0.625 µM) or with DMSO (vehicle control). We then isolated apoptotic cells using fluorescence activated cell sorting (FACS) with an antibody specific for cleaved PARP and a fluorescence-conjugated secondary antibody. We isolated genomic DNA and identified shRNAs present in this apoptotic population by massively parallel sequencing. To rank the shRNAs by their relative abundance in the PI3Ki-treated samples compared to DMSO-treated samples, we used a T-score to account for both the magnitude of the difference in means and the consistency of replicates. We then used the STARS (*Doench et al., 2016*) and RIGER (*Luo et al., 2008*) algorithms, which require that at least two

shRNAs targeting a gene are ranked significantly higher than random, to rank genes targeted by the top scoring shRNAs. Based on these two algorithms, 57 genes met this significance threshold (*Figure 1—figure supplement 1E*), of which 36 genes were called by both algorithms (*Figure 1—figure supplement 1F* and *Supplementary file 1*).

To confirm the 57 candidates, we used a GFP-based cell proliferation assay in which shRNAs were introduced into GFP-expressing cells, and fluorescence was used as a surrogate for cell number both prior to and after 9 days of treatment. The fluorescent measurements were then used to calculate the fold-change of proliferation (FC-proliferation), where a value of 1 corresponds to proliferative arrest, and cell death results in FC-proliferation <<1. Cells infected with control shRNAs and treated with GDC0941 exhibited FC-proliferation ≈ 1 and were used to calculate Z-scores for the FC-proliferation of experimental shRNAs, with a Z-score <<0 corresponding to cell death. We first retested the shRNAs that scored in the screen and then performed a second level validation by testing additional shRNAs targeting these genes for their ability to facilitate cell death upon PI3K inhibition (*Supplementary file 2*). In parallel, we also measured the knockdown efficiency of all shRNAs by qPCR. We used two criteria to select genes for further study: (i) their expression was suppressed by three or more targeting shRNAs (*Figure 1—figure supplement 1G*) and, (ii) their suppression by at least three targeting shRNAs modified the response to PI3K inhibition (*Figure 1B*). Using these criteria, we confirmed five candidate genes: *PIM2, TACC1, ZAK, ZFR, and ZNF565*. In addition, to determine whether suppression of these candidates affected PI3K signaling, we assessed the consequences of suppressing these genes on AKT phosphorylation (*Figure 1—figure supplement 1H*) and found that suppressing none of these candidates reduced phosphorylation at Ser473 of AKT, indicating that the observed interaction with PI3K inhibition was not due to further inhibition of the pathway.

To further validate the interaction of the five identified genes with PI3K inhibition, we measured two features characteristic of apoptotic cells. First, we measured PARP cleavage as an indicator of the induction of apoptosis (*Figure 1C*). We found that cells infected with control shRNAs (EmptyT or Luciferase) and treated with GDC0941 did not exhibit cleaved PARP. In contrast, every shRNAs targeting the identified genes induced PARP cleavage following treatment with GDC0941, indicating the induction of apoptosis. Notably, although suppression of each of these genes resulted in reduction in the proliferation rate of untreated cells by 2–4 fold as compared to control shRNAs, we did not observe evidence of apoptosis as measured by cleaved PARP (*Figure 1D*), indicating that the screen identified genes that only affect cell survival in the presence of the PI3K inhibitor. In addition, we used flow cytometry to examine the cell cycle profile to assess the proportion of cells with a sub-G1 DNA content, as a second measure of apoptosis (*Figure 1E*). In consonance with the cleaved PARP observations, suppression of each of the five identified genes resulted in an increased percentage of cells with a sub-G1 DNA content in the presence of GDC0941 compared to control cells. Taken together, we identified and validated five genes that convert the cytostatic PI3K inhibitory response into apoptotic cell death. As indicated above, the observed effects were obtained using at least three different shRNAs containing different seed sequences, which ensured that these effects were the consequence of on-target activity of the shRNAs, rather than off-target suppression due to miRNA effects mediated by seed sequences (*Jackson et al., 2006*).

## Validating candidates

In addition to using multiple shRNAs per target genes, we used two additional strategies to validate on-target effects. First, we expressed the corresponding ORF to rescue the effect of each shRNAs. To prevent the suppression of exogenously expressed ORFs by shRNAs, we introduced at least three synonymous mutations into the shRNA-targeted sequence (*Figure 2—figure supplement 1A*). MDA-MB-453 cells were infected with lentivirally-delivered V5-tagged ORFs, followed by infection with lentivirally-delivered shRNAs. Cells were then treated with DMSO or GDC0941 for 9 days. We used In-Cell-Western to detect expression of the V5 tag as a measure of the ORF-infected cell population. Over-expression of *PIM2, ZNF565*, and *TACC1* rescued the effect of endogenous gene suppression by their corresponding shRNAs (*Figure 2A*). We also found that over-expression of the kinase-inactive mutant of PIM2 (K61A) failed to rescue the suppression of endogenous *PIM2*, suggesting that the kinase activity of PIM2 is essential for preventing cell death following PI3K inhibition. Overexpression of *ZAK* reduced cell viability in a kinase-dependent manner, but independent of shRNA suppression or PI3K inhibition (*Figure 2—figure supplement 1B*), suggesting that *ZAK*

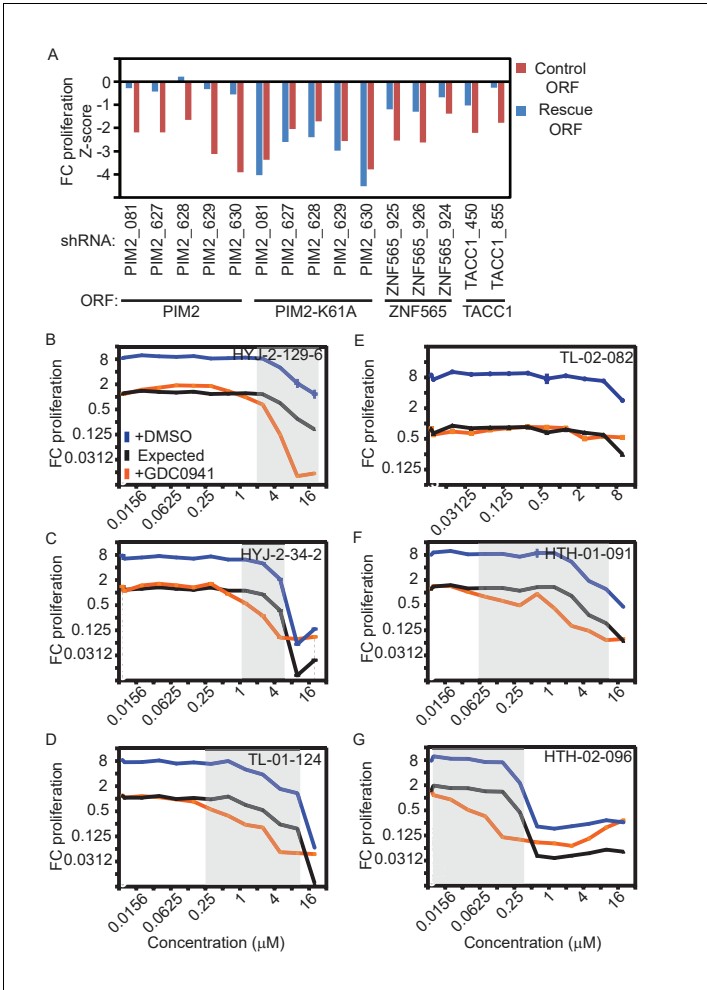

**Figure 2.** Validation of genes identified in the screen. (**A**) MDA-MB-453 cells were infected with the indicated ORFs and shRNAs. Cells were then treated with GDC0941 (0.625 µM) for 9 days, followed by detection of V5-expressing cells by ICW. Cells infected with five different control shRNAs were used to calculate Z-scores. Data shown are representative of three independent experiments. (**B–G**) MDA-MB-453 cells were treated for 9 days with the indicated inhibitors (0.01–20 µM) combined with GDC0941 (0.625 µM; red), or vehicle (DMSO; blue). Plotted are mean and standard deviation of fold-change proliferation from quadruplicates. The expected fold-change of proliferation for treatment combination (black) was calculated according to the Bliss independence model using single treatment effects. The gray shading indicates compound concentrations for which significant synergy with GDC0941 was observed. Data shown are representative of two independent experiments.

The following figure supplement is available for figure 2:

**Figure supplement 1.** Supporting information for ORF rescue validation.

overexpression itself reduces cell fitness. We found that overexpression of *ZFR* failed to rescue *ZFR* suppression (*Figure 2—figure supplement 1C*). To investigate potential interactions between the five candidate genes, we utilized the rescue assay described here to test whether any of the genes can be rescued by another gene. These experiments revealed that *PIM2* and *TACC1* over-expression partially interfered in the interaction between PI3K inhibition and suppression of the other genes (*Figure 2—figure supplement 1D*), suggesting that PIM2 or TACC1 generally contribute to cell survival in the presence of PI3K inhibition. Taken together, ORF expression-mediated rescue further validated three of the five identified genes.

## Small molecule inhibitors of PIM2 and ZAK synergize with GDC0941

A second strategy to confirm on-target effects of shRNAs is to functionally mimic target suppression by using small molecule inhibitors. PIM2 and ZAK are serine/threonine kinases, therefore are potential targets for small molecule inhibitors. Searching for compounds that exhibit a selective and potent binding to either PIM2 or ZAK, we identified two compounds that target PIM2 and three compounds targeting ZAK (*Supplementary file 3*). Specifically, PIM2 scored high in a kinase selectivity assay for two compounds, HTH-01–091, and HTH-02–096, which also exhibited high potency for PIM2 inhibition in a biochemical assay (*Supplementary file 3*). Of note, the potency of both of these compounds as PIM2 inhibitors is similar to their potency toward PIM1, in contrast to commercially available PIM inhibitors that have a strong preference for PIM1 inhibition. Similarly, ZAK scored high in a focused kinase selectivity assay for one compound (TL-01–124). In a biochemical assay to determine IC50, three compounds exhibited high potency as ZAK inhibitors (TL-01–124, HYJ-2-129-6 and HYJ-2-34-2).

We used these small molecule inhibitors to further test the capacity of PIM2 or ZAK inhibition to modify the response to PI3K inhibition. We treated MDA-MB-453 cells with either DMSO or GDC0941 (0.625 µM) in combination with each of the several compounds targeting PIM2 and ZAK, and GFP fluorescence was used to determine changes in proliferation. We then used the Bliss independence model to determine synergy based on the expected versus the observed effect for each combination. When significant synergy occurs, as indicated by gray shading (*Figure 2B–G*; Bliss index <0.6), the observed effect (red) is greater than the expected effect (black). At higher concentrations, the calculated expected combined effect is limited by population death, which makes it impossible to determine whether synergy exists at these points. Combining inhibitors targeting ZAK (HYJ-2-129-6, HYJ-2-34-2, and TL-01–124) and PI3K resulted in a synergistic response and conversion of the GDC0941-induced proliferation arrest into cell death (*Figure 2B–D*). We also found that TL-02–082, an analog of TL-01–124 that does not bind ZAK, but otherwise has a similar kinase selectivity profile, did not exhibit a synergistic interaction with GDC0941 (*Figure 2E*). Similarly, co-inhibition of PIM2 (HTH-01–091, and HTH-02–096) together with PI3K also resulted in a synergistic response and conversion into cell death (*Figure 2F–G*). These observations further validate that PIM2 and ZAK inhibition synergizes with PI3K inhibition to induce cell death.

## Confirming candidates in breast cancer cell lines

To confirm the relevance of the candidates in other cell lines, we suppressed these same five genes in three additional breast cancer cell lines, including HCC1954, T47D, and HCC1937, which growth arrest upon PI3K inhibition. Each of these cell lines show activation of the PI3K pathway through different mechanisms: *ERBB2* amplification (HCC1954), *PIK3CA* amplification (HCC1937) or activating mutation (T47D, HCC1954), and *PTEN* loss (HCC1937). We introduced shRNAs targeting the five genes into these cell lines and monitored their proliferation (*Figure 3A*). Similar to the effects observed in MDA-MB-453 cells, we found that suppression of these five genes in each of these cell lines lead to apoptosis in the setting of PI3K inhibition. In contrast to the proliferation inhibitory effect of PI3K inhibition on breast cancer cell lines with activated PI3K pathway, inhibition of PI3K in breast cancer cell lines that have a wild-type PI3K pathway (CAL-120 and MDA-MB-231) failed to affect cell proliferation. In these cell lines, suppression of the five genes had no effect on cell proliferation or apoptosis when combined with PI3K inhibition. We also found that although PI3K inhibition in cells with a wild-type PI3K pathway activated by growth factors can suppress proliferation, suppression of these five genes did not modify this response (*Figure 3A*; MCF10A). In addition to breast cancer cell lines, we also tested four additional non-breast cancer cell lines. The ovarian cancer cell line EFO-21 (*PIK3CA* amplified) exhibited similar response to that observed in breast cancer cell lines with oncogenic PI3K activation (*Figure 3—figure supplement 1A*). However, the interactions we observed between PI3K inhibition and suppression of the five candidate genes in breast cancer cell lines did not occur in the three tested GBM cell lines (all that harbor PTEN inactivation) (*Figure 3—figure supplement 1B*). Taken together, these results suggest that the response to PI3K inhibition can be modified by the suppression of the five identified genes only in the context of oncogenic PI3K pathway activation.

In addition, we found that the combination of ZAK or PIM2 inhibition and PI3K inhibition also induced cell death at a synergistic manner in four additional cell lines: T47D, HCC1954, HCC1937,

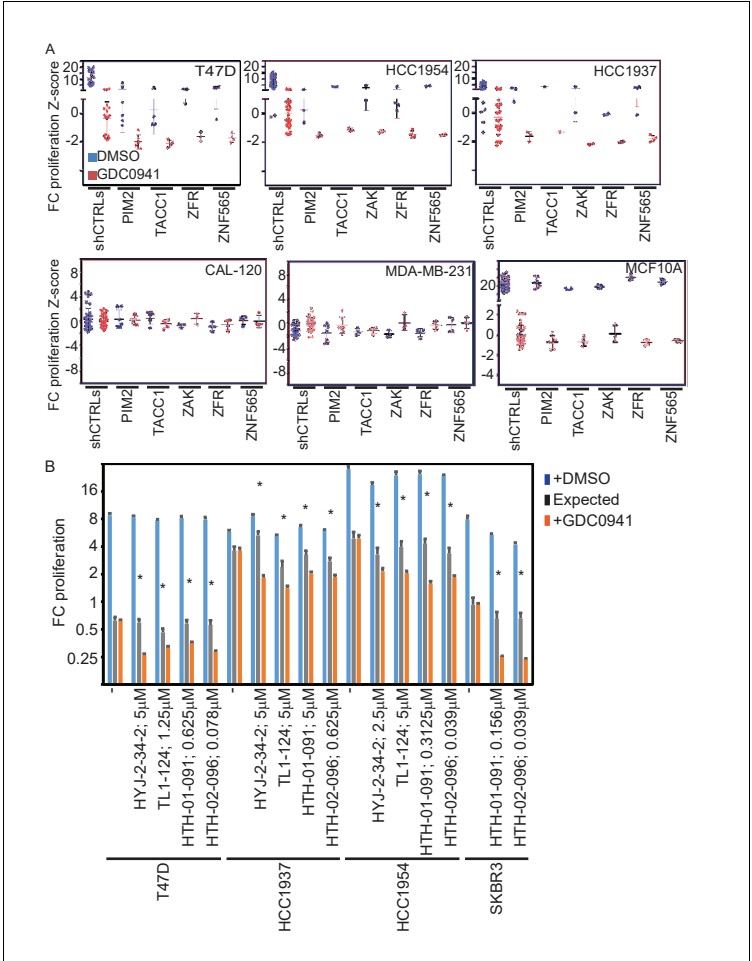

**Figure 3.** Effects of manipulating candidate gene expression in multiple cell lines. (**A**) T47D-eGFP, HCC1954-eGFP, HCC1937-eGFP, CAL-120-eGPF, MDA-MB-231-eGPF, and MCF10A-eGPF were infected with shRNAs targeting the indicated genes. Cells were then treated with GDC0941 (0.5, 1.25, 1, 0.312, 0.312, and 0.156 µM, respectively) or vehicle (DMSO) for 9 days. Z-scores for the fold-change of proliferation were calculated based on control shRNAs. Bars indicate standard deviation between the different shRNAs targeting each gene. Data shown are representative of at least two independent experiments. (**B**) T47D-eGFP, HCC1954-eGFP, HCC1937-eGFP, and SKBR3-eGFP were treated for 9 days with the indicated inhibitors (0.01–10 µM) combined with GDC0941 (0.5, 1.25, 1, and 0.625 µM, respectively; red), or vehicle (DMSO; blue). Plotted are mean and standard deviation of fold-change of proliferation from 4 replicates. Expected effect of combination treatment (gray) was calculated as in *Figure 2*. Significant synergistic combinations according to Bliss independence model are indicated by asterisks.

The following figure supplement is available for figure 3:

**Figure supplement 1.** Effects of manipulating candidate genes in non-breast cancer cell lines.

---

and SKBR3 (bearing *PTEN* deletion and *ERBB2* amplification) (*Figure 3B*). These observations confirm that suppression of these 5 genes converts PI3Ki-mediated proliferative arrest into a cytotoxic response in multiple cell lines with oncogenic PI3K pathway activation.

## Specificity of the interactions between PI3K inhibition and the candidates

To explore the pathway specificity of these observed interactions, we tested whether the identified genes also modify the response to up- or downstream inhibition of the PI3K pathway. Following the transduction of the shRNAs, MDA-MB-453 cells were treated with either a HER2 inhibitor

(BIBW2992, 1 µM) or an AKT inhibitor (MK2206, 0.5 µM) at doses that induce significant attenuation of cell proliferation. Suppression of all five genes modified the response to HER2 or AKT inhibition, similar to our observation with PIK3CA inhibition (*Figure 4A*). Similarly, suppression of the five identified genes combined with inhibition of PDK1 (GSK2334470, 2.5 µM) or mTOR (Sirolimus, 10 nM) modified the response to PDK1 or mTOR inhibitors (*Figure 4—figure supplement 1A–B*). These observations demonstrate that the observed synergy between these genes and PI3K pathway

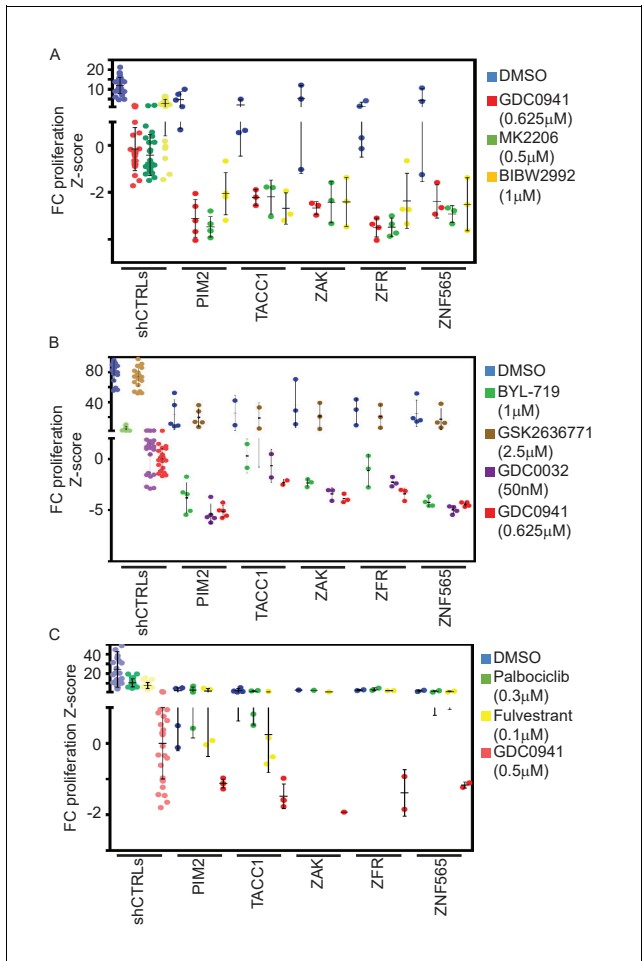

**Figure 4.** PI3K pathway specificity of identified interactions. (**A**) Z-scores for fold-change of proliferation of MDA-MB-453-eGFP cells infected with shRNAs targeting the indicated genes and treated for 9 days with GDC0941 (0.625 µM; red), MK2206 (0.5 µM; green), BIBW2992 (1 µM; yellow) or vehicle (DMSO; blue). Cells infected with five different control shRNAs were used to calculate Z-scores. Bars indicate standard deviation between the different shRNAs targeting each gene. Data shown are representative of two independent experiments. (**B**) Z-scores for fold-change of proliferation of MDA-MB-453-eGFP cells infected with shRNAs targeting the indicated genes and treated for 9 days with GDC0941 (0.625 µM; red), BYL-719 (1 µM; green), GSK2636771 (2.5 µM; brown), GDC0032 (50 nM; purple) or vehicle (DMSO; blue). Cells infected with five different control shRNAs were used to calculate Z-scores. Bars indicate standard deviation between the different shRNAs targeting each gene. Data shown are representative of three independent experiments. (**C**) Z-scores for fold-change of proliferation of T47D-eGFP cells infected with shRNAs targeting the indicated genes and treated for 9 days with GDC0941 (0.5 µM; red), Palbociclib (0.3 µM; green), Fulvestrant (0.1 µM; yellow) or vehicle (DMSO; blue). Cells infected with five different control shRNAs were used to calculate Z-scores. Bars indicate standard deviation between the different shRNAs targeting each gene. Data shown are representative of three independent experiments.

The following figure supplement is available for figure 4:

**Figure supplement 1.** Validation of pathway specificity with additional PI3K-pathway inhibitors.

inhibition is not specific to the particular PI3K inhibitor (GDC0941) used in the screen. To further explore the specificity of PI3K inhibition, we tested isoform specific inhibitors of PI3K, to inhibit either the alpha (BYL-719) or the beta (GSK2636771) isoforms. In addition, we also tested the beta-sparing PI3K inhibitor GDC0032. As shown in *Figure 4B*, the only inhibitor that failed to interact with the suppression of the five identified genes is the beta isoform inhibitor, thus suggesting that the alpha isoform, PIK3CA, is the major isoform that mediates the observed interactions.

Since PI3K inhibition attenuates proliferation in these cells, it is possible that the observed modified response to PI3K inhibitors by these five genes is the result of suppressing these genes in a state of impaired proliferation rather than being specifically related to PI3K inhibition. To determine whether this was the case, we combined shRNA-mediated suppression of these candidate genes with two PI3K-independent anti-proliferative drugs, Palbociclib, a CDK4/6 inhibitor, and Fulvestrant, an ER antagonist. In contrast to what we observed following PI3K inhibition, suppression of the five genes did not convert the cytostatic effect of either CDK4/6 inhibitor or ER antagonist into cytotoxicity (*Figure 4C*). These observations indicate that the genes that we identified in this screen are specific modifiers of PI3K pathway inhibition and do not non-specifically affect cells whose proliferation is attenuated by other mechanisms.

## PIM2 suppression increases mitochondrial priming

BAD, a BH3-only protein whose phosphorylation leads to its sequestration by 14-3-3 proteins, is a substrate of PIM2 (*Danial, 2008*; *Yan et al., 2003*). We hypothesized that BAD phosphorylation mediates the effect of PIM2 on PI3K inhibition. To test this hypothesis, we first confirmed that BAD phosphorylation was reduced following *PIM2* suppression with either shRNAs or a small molecule

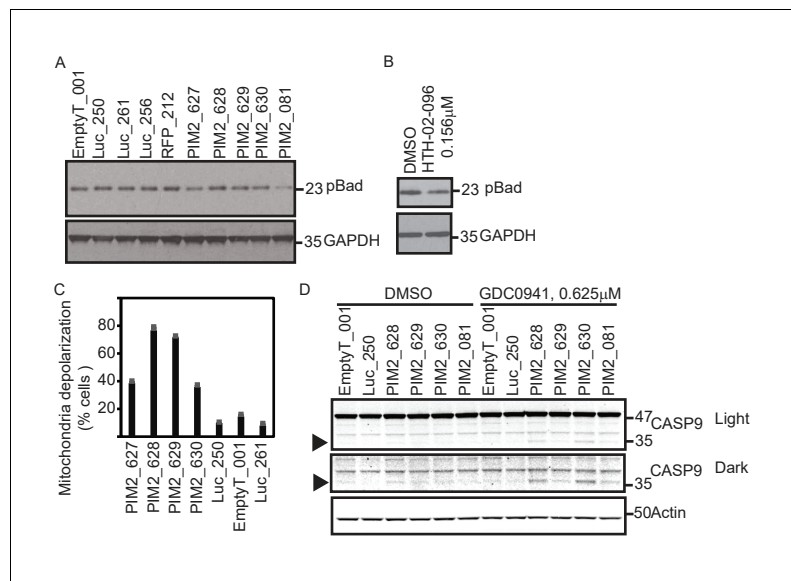

**Figure 5.** PIM2 suppression increases mitochondrial priming.  (**A**) MDA-MB-453 cells were infected with the indicated shRNAs. Five days after infection, cells were collected and phosphorylation of Bad was detected by Western blotting. (**B**) MDA-MB-453 cells were treated with PIM2 inhibitor HTH-02–096 for 24 hr, and Bad phosphorylation was detected as in A. (**C**) MDA-MB-453 cells were infected with the indicated shRNAs. Five days after infection, cells were collected and subjected to BH3 profiling. The percentage of cells exhibiting mitochondrial depolarization at 0.4 µM BIM peptide is plotted. Bars indicate standard deviation of 3 replicates. Data shown are representative of three independent experiments. (**D**) MDA-MB-453 cells were infected with the indicated shRNAs, and then treated for 30 hr with GDC0941 (0.625 µM) or left untreated. Adherent and floating cells were collected and subjected to immunoblot analysis for caspase 9. Triangle marks the product of cleaved caspase 9.

The following figure supplement is available for figure 5:

**Figure supplement 1.** MDA-MB-453 cells were infected with the indicated shRNAs.

PIM2 inhibitor (*Figure 5A–B*). To test the consequences of reducing BAD phosphorylation by suppressing *PIM2*, we applied the BH3 profiling assay to assess whether *PIM2* suppression primed mitochondria for apoptosis (*Certo et al., 2006*). Mitochondrial priming level is determined by measuring the ratio between pro- and anti-apoptotic Bcl-2 proteins. BH3 profiling measures mitochondrial priming by determining whether synthetic peptides modeled after the BH3 domains of Bcl-2 proteins induce loss of cytochrome c from mitochondria, which indicates the likelihood of cells to undergo apoptosis and correlates with patient response to chemotherapy (*Ni Chonghaile et al., 2011*). Suppression of *PIM2* significantly increased overall mitochondrial priming (*Figure 5C*), consistent with the observation that *PIM2* suppression reduces the phosphorylation of BAD. In contrast, suppression of any of the other four genes did not lead to an increase in mitochondrial priming level (*Figure 5—figure supplement 1*), suggesting that the mechanism of interaction between PIM2 and PI3K inhibition is unique among the five identified genes. Supporting this mechanism of interaction, suppression of *PIM2* resulted in caspase 9 cleavage (*Figure 5D*) when combined with PI3K inhibition, indicating the involvement of mitochondria depolarization in the observed cell death.

## Suppression of the identified genes modifies the response to PI3K inhibition in vivo

To determine whether the synergy between suppression or inhibition of the identified genes and PI3K inhibition was relevant for tumor maintenance in vivo, we used an implantable microscale device for simultaneous intratumoral delivery of several siRNAs or small molecules directly into xenografts of MDA-MB-453 cells (*Jonas et al., 2015*). The major advantage of this device is that it allows more precise delivery of siRNA or small molecules to the tumor, allows for better quantification of responses and facilitates the testing of combinations of treatments. Each of 16 reservoirs on the device was loaded with ~1 µg of a unique siRNA or a small molecule. The device allows compounds to be released into confined regions within the tumor over 48 hr for siRNA and for 24 hr for small molecules, respectively. The device was then removed along with a cylindrical column of surrounding tumor tissue, which was stained by IHC and analyzed to determine cellular outcome. To facilitate efficient delivery, siRNAs were Accell-modified (Dharmacon, GE Healthcare Life Sciences) and labeled at the 5'-end with a fluorescent Cy3 dye to allow for detection of siRNA delivery into tissue (*Figure 6A*).

To assess the effect of combining suppression of our five genes with PI3K inhibition, mice harboring xenograft tumors of MDA-MB-453 cells were systemically dosed with GDC0941 daily for 5 days prior to device implantation and during the period until the device was removed. Tissue surrounding the device was stained with hematoxylin and eosin and the density of intact nuclei around each reservoir was quantified as a measure of cell viability. Combining PI3K inhibition with control siGFP did not affect cell density compared to the effect of siGFP on tumor cells from non-treated mice. However, combining PI3K inhibition with any of the siRNAs targeting the five identified genes lead to a significant reduction in the density of intact nuclei as compared to non-treated control mice (*Figure 6—figure supplement 1B–C*). For all siRNAs targeting any of the five candidate genes, this reduction in density of intact nuclei upon combination treatment was statistically significant ($p<0.05$) when compared to either single treatments. We calculated the proportional effect from non-treated control siRNA (siGFP) for single treatments (each of the siRNAs or PI3K inhibition alone) and combined treatments, and compared it to the expected effect according to the Bliss independence model. All siRNAs targeting the five identified genes combined with PI3K inhibition exhibited greater effect then expected with a Bliss value significantly smaller than 1 (*Figure 6B*). Thus, siRNA suppression of each of the five identified genes combined with PI3K inhibition lead to a significant increase in cell death (*Figure 6B* and *Figure 6—figure supplement 1A*).

Similarly, we also tested the combined effect of small molecule ZAK inhibitors and PI3K inhibition, and found that the combination resulted in a significant increase in the percentage of apoptotic cells as evidenced by staining for cleaved-CASP3 (*Figure 6C* and *Figure 6—figure supplement 1B*). Notably, loading doxorubicin into a reservoir of the device lead to cleaved-CASP3 staining in ~20% of the exposed cells in tissue, which was not further affected by PI3K inhibition. This observation indicated that the observed increase in cleaved-CAPS3 with ZAK and PI3K inhibitors reflects a specific synergistic interaction. Taken together, these studies confirm that suppression or inhibition of the genes identified herein show synergistic effects with PI3K inhibition on tumor survival in vivo.

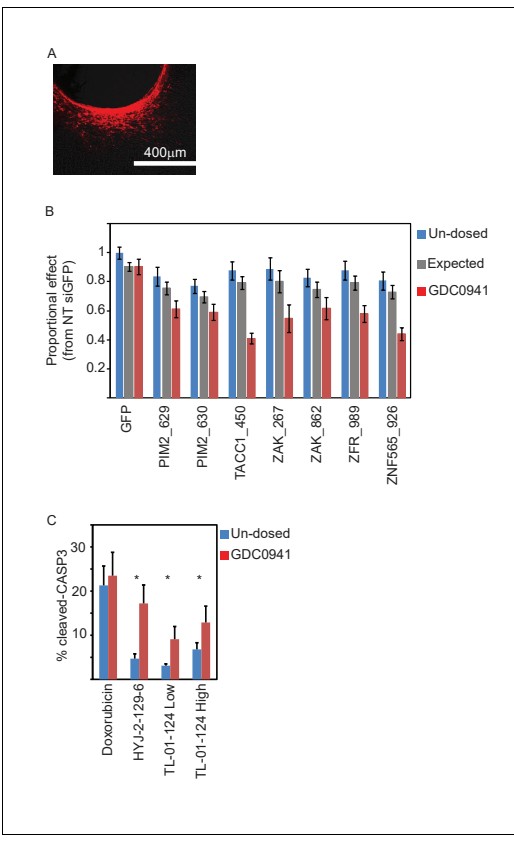

**Figure 6.** In vivo validation of the five identified genes. (**A**) A representative image showing siRNA (red) spreading in tumor tissue. (**B**) Devices loaded with the indicated siRNAs were implanted into MDA-MB-453 xenografts of un-dosed mice (blue) or mice that were pre-dosed with GDC0941 (75 mg/kg) for 5 days (red). Two days later, the devices were removed, xenograft tissue surrounding each reservoir was stained for Hematoxylin and Eosin, and the density of intact nuclei was quantified. Plotted are the proportional effects of the indicated siRNAs with or without PI3K inhibition as compared to the effect of siGFP in non-treated mice. Bars indicate SD from at least 4 replicates. The expected proportional effect for treatment combination (grey) was calculated according to the Bliss independence model using single treatment effects. (**C**) Devices were loaded with the indicated small molecules and were implanted into MDA-MB-453 xenografts of un-dosed mice (blue) or mice that were pre-dosed with GDC0941 (75 mg/kg) for 5 days (red). 24 hr later, the devices were removed, xenograft tissue surrounding each reservoir was stained for hematoxylin and eosin and for cleaved-CASP3. Percentage of cells stained positive for cleaved caspase 3 was quantified. Mean and SD of at least 6 replicates is shown, asterisks indicate T-test p-value<0.05.

The following figure supplement is available for figure 6:

*Figure 6 continued on next page*

## Discussion

Here we report the identification of genes that when suppressed in the setting of a PI3K inhibitor induce cell death. Specifically, we identified five genes, *PIM2, TACC1, ZAK, ZFR*, and *ZNF565* whose suppression converts the cell cycle arrest induced by PI3K inhibition into apoptosis (*Figure 1*). Of these five genes, two are kinases for which we identified small molecule compounds that recapitulated the effects observed by shRNAs (*Figure 2*). In addition to using multiple distinct shRNAs targeting each of the five genes, we independently confirmed the initial screen using either small molecule inhibitors (for PIM2 and ZAK) or a rescue assay (*PIM2, TACC1*, and *ZNF565*) (*Figure 2*). Although we were unable to independently confirm *ZFR* by either CRISPR-CAS9-mediated gene deletion or by a rescue assay, we included *ZFR* suppression in the experiments herein to provide information on this candidate for future studies. Our observations suggest that co-targeting PI3K and either PIM2 or ZAK is a synergistic combination with therapeutic potential. Furthermore, using a microscale implementable device, we found that inhibition or suppression of these genes also synergizes with PI3K inhibition in vivo (*Figure 6*). Together these studies identify potential targets whose perturbation augments the effects of PI3K inhibition.

Although the primary screen described here was conducted in a single cell line, we verified these interactions in additional breast cancer cell lines in which the PI3K pathway is activated by different oncogenic aberrations (*Figure 3*). We also showed that *PIK3CA* is the major isoform that mediates this interaction (*Figure 4A–B*). Importantly, these genes did not modulate the effect of drugs unrelated to the PI3K pathway, such as a CDK4/6 inhibitor and an ER antagonist (*Figure 4C*). We also note that the five genes failed to interact with PI3K inhibition in cells that lack oncogenic activation of the pathway (*Figure 3*). These observations serve to further define the context in which these targets could be therapeutically combined.

Although we identified and validated several candidates, we recognize that this screen was not saturating due to technical shortcomings of the shRNA library and inefficiencies of flow cytometry based assay that we employed. Thus, although we identified genes whose suppression in the presence of a PI3K inhibitor induced apoptotic cell death, extending this approach is likely to identify additional candidates that when

*Figure 6 continued*

**Figure supplement 1.** Supporting information for in vivo validation.

suppressed or inhibited synergize with PI3K inhibition.

A concern in drug-modifier screens is the potential for false positive results due to a combined effect on cell fitness rather than a specific functional interaction. In such cases, each perturbation independently reduces the cellular fitness and when combined, the effect on cellular fitness leads to cell death. One would expect that alternate combinations with other growth-arresting perturbations would have similar cytotoxic consequences. Here we demonstrated that the five identified genes from our screen caused cell death specifically with PI3K pathway inhibition and not with other cytostatic perturbations (*Figure 4*), suggesting that these interactions are specific for this genetic and pharmacologic context.

We previously showed that over-expression of *PIM1* can similarly rescue cells from lapatinib-induced apoptosis (*Moody et al., 2015*), corroborating a role for *PIM2* in promoting resistance to PI3K inhibition. In addition, it has been shown that PIM and AKT inhibitors synergize in acute myeloid leukemia (*Meja et al., 2014*). PIM2 promotes tumor survival mainly through phosphorylation of the pro-apoptotic BH3-only protein BAD, as well as through regulation of MYC activity, and Cap-dependent protein translation (*Nawijn et al., 2011*). Because BAD phosphorylation is a major pro-survival mechanism of PIM2, we tested and confirmed that PIM2 suppression leads to mitochondrial priming (*Figure 5*). However, it is likely that additional substrates of *PIM2* contribute to the interaction with PI3K inhibition. Of note, AKT and the PIM family of proteins share several common substrates, including BAD, while also having a number of unique targets (*Amaravadi and Thompson, 2005*; *Nawijn et al., 2011*). Additional kinases such as PKA and p90RSK, also regulate BAD by phosphorylation (*Danial, 2008*). Accordingly, BAD phosphorylation was shown previously to integrate survival signals from the PI3K and MAPK pathways (*She et al., 2005*).

One of the other candidates identified in this screen, *TACC1*, has been shown to enhance mammary tumor formation induced by heterozygous *PTEN* loss in vivo and was shown to mediate cell survival following PI3K pathway inhibition induced by *PTEN* over expression (*Cully et al., 2005*). TACC1 interacts with Aurora A-chTOG complex at the spindle pole during metaphase (*Conte et al., 2003*). At later stages of M phase, during anaphase and cytokinesis, TACC1 localizes to the midzone spindle and the midbody together with Aurora B (*Delaval et al., 2004*). Suppression of *TACC1* leads to formation of multipolar spindles and thus perturbs cell division (*Conte et al., 2003*). Notably, PI3K activity contributes to genomic stability by regulating spindle orientation and chromosomal segregation (*Silió et al., 2012*). These converging mechanisms suggest that combined suppression of TACC1 and PI3K may induce cell death by causing un-resolved spindle and chromosomal segregation defects.

ZAK is a MAPK kinase kinase (MAP3K) that transduces signals through JNK and p38, and is associated with activation of AP1 and NF-κB (*Liu et al., 2014*). *ZAK* is up-regulated in several cancers including breast cancer (*Liu et al., 2014*). *ZAK* over-expression has been shown to promote in vivo transformation (*Cho et al., 2004*) and cell migration (*Rey et al., 2016*), and its suppression reduced proliferation (*Liu et al., 2014*) and β-catenin activity (*Firestein et al., 2008*). Growth-factor stimulation, specifically EGF stimulation, activates ZAK (*Cho et al., 2004*; *Rey et al., 2016*), which then contributes to the activation of ERK signaling (*Vinayagam et al., 2011*). In contrast, it was recently shown that ZAK inhibition by Sorafinib causes cutaneous squamous cell carcinoma due to suppression of JNK activity (*Vin et al., 2014*), which might explain why over-expression of exogenous *ZAK* lead to reduced viability. Taken together, these observations suggest that ZAK influences cell state in a context-dependent manner, and that ZAK can either promote or suppress cell proliferation. Here we have demonstrated that *ZAK* suppression induced cell death upon PI3K inhibition, indicating that ZAK-mediated signaling is essential for cell survival in the context of PI3K inhibition in breast cancer cell lines.

The role of *ZFR* and *ZNF565* in cellular proliferation and survival, especially in cancer and in the context of PI3K pathway inhibition, are not yet known. Further studies will be necessary to investigate the interaction of ZFR and ZNF565 with PI3K inhibition.

In consonance with the known pro-survival function of PIM2 and TACC1 in the context of PI3K inhibition, we show here that their over-expression partially rescued the effect of combining PI3K

inhibition with the other four candidate genes (*Figure 2—figure supplement 1D*), further emphasizing the importance of combining their inhibition with PI3K inhibitors.

Although there is no reason to expect that these 5 genes would be altered in human cancers, analysis of breast cancer genome sequencing data collected by the TCGA project (http://www.cbio-portal.org) reveals that the five genes are either amplified or their expression is up-regulated in 37% of breast cancer patients. In this dataset, we failed to find co-occurrence or mutual exclusivity between any of the five genes. Samples sub-classified as PI3K pathway active or inactive (classification was based on *PIK3CA* mutations and amplification, *ERBB2* amplification or over-expression, or *PTEN* deletion or mutation) exhibited similar rates of these alterations, regardless of *PIK3CA* mutation status or breast cancer subtype. Moreover, there is no information regarding the response of the tumors analyzed by TCGA to PI3K pathway inhibition. As data from patients treated with PI3K inhibitors accumulates, it would be important to explore whether these genes are associated with response rate, or resistance.

In summary, we report the identification of five genes whose suppression both in vitro and in vivo modulates the cellular response to PI3K inhibition, converting cytostatic effects into cytotoxicity. Importantly, we report that small molecule inhibition of two of these genes, *PIM2* and *ZAK*, is synergistic with PI3K inhibition, a finding that has therapeutic implications for the treatment of breast cancer.

## Materials and methods

### Cell lines

MDA-MB-453 (RRID:CVCL_0418), SKBR3 (RRID:CVCL_0033), MDA-MB-231 (RRID:CVCL_0062), CAL-120 (RRID:CVCL_1104), LN-443 (RRID:CVCL_3960), LN-382 (RRID:CVCL_3956), and SF-295 (RRID:CVCL_1690) were grown in DMEM supplemented with 10% FBS. T47D (RRID:CVCL_0553), HCC1954 (RRID:CVCL_1259), HCC1937 (RRID:CVCL_0290), and EFO-21 (RRID:CVCL_0029) were grown in RPMI supplemented with 10% FBS. MCF10A (RRID:CVCL_0598) cells were grown in MEGM supplemented with Bullet kit (Lonza, USA). To generate eGFP-expressing cell lines, cells were infected with a lentivirally-delivered eGFP- and blasticidin-encoding plasmid. Cells were selected with blasticidin for at least 7 days. Unless specified, all media and supplements were from Gibco (Thermo Fisher Scientific, USA). All drugs were from Selleck Chemicals (USA). All cancer cell lines were obtained from the Cancer Cell Line Encyclopedia, which obtained them directly from original sources and confirmed their identity by SNP fingerprinting (*Barretina et al., 2012*). LN-443 is included in the list of commonly misidentified cell lines (International Cell Line Authentication Committee) and is considered to be LN-444. Both harbor the same PTEN inactivating mutation and therefore for the purpose of this report it is considered as PTEN inactive. MCF10A cells were obtained from ATCC (ATCC, USA). All cells were tested for mycoplasma periodically as well as prior to screening.

### Virus production and infection

Lentivirus was produced in 293 T cells by co-transfection with VSV-G envelope encoding plasmid, psPAX2 packaging plasmid, and pLKO or pLEX viral vector. Virus-containing media were harvested 48 hr post transfection. For a single virus infection, cells were plated 24 hr prior to infection. Virus-containing media and polybrene (8 ug/ml final concentration) were added to cells, after which cells were centrifuged at 1200 RPM for 45 min to promote infection. Virus-containing media were replaced 24 hr post infection.

### Cell proliferation assay

eGFP-expressing cells were seeded in phenol red-free media in 384 well plates using a Multidrop Combi reagent dispenser (Thermo Scientific, USA). Infection and/or drug treatment were done in quadruplicates to obtain technical replicates. At least five different control shRNAs that do not target an endogenous transcript were used, and each control shRNA was infected into at least four quadruplicates across the plate, so that at least 20 quadruplicates were infected with control shRNAs per plate. Fluorescence intensity was measured by a SpectraMax M5 (Molecular Devices, USA) immediately after initial addition of GDC0941 (day 0) and after 9 days of treatment (day 9). Media

and drugs were replaced every 3–4 days. Background fluorescence from media-only wells was subtracted from fluorescent values of all cell-containing wells. Fold-change in proliferation was calculated by dividing the mean fluorescence on day nine by the mean fluorescence on day 0 for each quadruplicate. The mean FC-proliferation of all control shRNAs was used to calculate Z-scores for the FC-proliferation of experimental shRNAs.

## FACS

Floating and adherent cells were collected and fixed over-night in either 100% cold methanol (for cell-cycle analysis), or in 70% cold ethanol (for cPARP staining). For cell cycle analysis, cells were rehydrated in PBS for 30 min, followed by staining with Propidium Iodide (25 ug/ml)/ RNAse A (50 ug/ml). Staining with anti-cPARP antibody (Cell Signaling Technology, #5625) was conducted according to manufacturer's recommendations. A BDBiosciences LSR II flow cytometer was used to acquire samples. At least 30,000 cells were acquired per sample. FlowJo software was used for analysis. A BDBiosciences FACSAria instrument was used for cell sorting.

## Rescue assay

In order to rescue shRNA knock down, at least three silent mutations were introduced in the shRNA-targeted sequence of each ORF. ORFs were cloned into a Gateway expression vector fused to a C-terminal V5 tag. For control ORF, HcRed was cloned into the same expression vector. Cells were seeded in 384 well plates using a Multidrop Combi reagent dispenser. Twenty-four hours later, cells were infected with lentivirally-delivered ORF vectors. 48 hr after ORF vector infection, cells were infected with lentivirally-delivered shRNAs. Infections were done in quadruplicates. Following a 48 hr recovery, cells were either treated with GDC0941 or DMSO for 9 days, with refreshing media and treatment every 3–4 days. At the end point, cells were fixed in 4% PFA, and stained for V5 (Invitrogen #46–0705) expression as a surrogate reporter for ORF expression, and DARQ5 DNA stain to assess total cell viability. Staining was detected and quantified using a LiCor Odyssey scanner (LiCor, USA) and Image Studio software. Z-scores were calculated as described for *cell proliferation assay*.

## Western blot analysis

Cells were collected and lysed in cold 1% NP-40 lysis buffer (50 mM Tris pH7.5, 150 mM NaCl, 2 mM EDTA, 1% NP-40, 1 mg/ml NaF) supplemented with phosphatase and protease inhibitors. Lysates were cleared by centrifugation, and protein concentration was determined by BCA assay. Samples were denatured by adding NuPage LDS sample buffer (Thermo Fisher Scientific, USA) and reducing agent and boiling for 5 min. Equal amounts of protein were separated by electrophoresis on a 4–12% Bis-Tris gel, and then transferred to a nitrocellulose membrane. Membranes were incubated with primary antibody over-night at 4°C, and then incubated with a secondary antibody conjugated to either IRDye or HRP. Signals were detected by either a LiCor Odyssey scanner, or by using an ECL kit (Perkin-Elmer, USA). All primary antibodies were from Cell Signaling Technology (USA).

## qPCR

Cellular RNA was purified with a kit (PerfectPure; 5 Prime GmbH, Germany) and complementary DNA (cDNA) was synthesized with the High Capacity RNA-to-cDNA kit (ABI, Thermo Fisher Scientific, USA). Real-time-qPCR analysis was performed with Power-SYBR Green (ABI, Thermo Fisher Scientific, USA) in a QuantStudio 6 RT-PCR machine (Thermo Fisher Scientific, USA). Three replicates were made for all qPCR assays. Probe-specific results were normalized to *GAPDH* RNA levels.

## Selectivity and biochemical assay for small molecule inhibitors

Kinase selectivity assays for PIM2 inhibitors were done by the International Center for Kinase Profiling (Dundee, UK). A KiNativ (ActivX Biosciences) (*Patricelli et al., 2011*) was used as a focused kinase selectivity assay for ZAK inhibitors. Potency of compounds was tested using a SelectScreen assay (Thermo Fisher Scientific, USA).

## BH3 profiling

BH3 profiling was done as described previously (*Ryan and Letai, 2013*). In brief, cells were permeabilized and incubated with increasing concentrations of synthetic BIM peptide (New England Peptide, USA). Cells were fixed and stained for endogenous cytochrome c (Biolegend, USA). Cells were analyzed by FACS (Fortessa BD Biosciences) to determine the population rate of peptide induced loss of cytochrome c. Assay was conducted in triplicates.

## Whole-genome shRNA screen

The initial screen was conducted by infecting 300 million cells with a pooled viral-shRNA library consisting of 98K shRNA plasmids, at a MOI of ~0.3, such that each individual shRNA is represented by approximately 1000 cells. This infection was carried out in four biological replicates. Cells were then selected with puromycin for 2 days, after which cells were expanded for an additional 3 days. Cells were then divided into five separate samples, four of which were treated with GDC0941 (0.625 µM), which were then collected daily at 48,72,96 and 120 hr. One sample was treated with vehicle control (DMSO) and was collected at 72 hr. Cells were fixed, stained for cPARP and sorted twice to achieve >95% purity of cPARP-positive population. Genomic DNA was extracted from isolated cells and was used to amplify the integrated shRNA, which was then sequenced by Illumina HiSeq (*Cowley et al., 2014*). Abundance of shRNAs in each sample was used to calculate a T-statistic per shRNA across all samples.

## In vivo validation

Animals were maintained under conditions approved by the Institutional Animal Care and Use Committee at the Dana-Farber Cancer Institute and at the Massachusetts Institute of Technology. MDA-MB-453 cells were injected subcutaneously into both flanks of female SCID mice (Taconic, USA) at a concentration of 10⁶ cells/flank in 50% Matrigel. Xenograft tumors were allowed to grow until reaching a diameter of ~5–8 mm. For GDC0941 dosing, mice were treated by a daily GDC0941 (75 mg/kg/day, in MCT) delivered by oral gavage of for 5 days prior to device implantation. For siRNA-loaded devices, mice received an additional dose of GDC0941 24 hr after device implantation. siRNAs (Dharmacon, USA) were Accell-modified, Cy3-conjugated at the 5' end, and processed for in-vivo use. Microscale devices for intratumor delivery of siRNA and small molecule compounds were manufactured as described in *Jonas et al., 2015*, and were implanted directly into the mouse xenograft tumor. Devices containing the small molecule compounds remained in situ for 24 hr, while siRNA-loaded devices remained in situ for 48 hr. The flank tumor was excised and the tissue containing the device was fixed for 24 hr in 10% formalin and perfused with paraffin. The specimen was sectioned using a standard microtome and tissue sections were collected from each reservoir level. Sections were stained with Hematoxylin and Eosin, as well as antibody-stained by standard immunohistochemistry using cleaved-caspase-3 antibody (Cell Signaling #9661) and scored using an ImageJ (v1.48) image analysis algorithm in a blinded manner.

## Acknowledgements

The authors would like to thank members of the Hahn lab and Dr. Ravid Straussman for fruitful discussions, and the International Centre for Kinase Profiling in University of Dundee. This work was supported in part by NIH grants U01 CA176058 and R01 CA130988. YZ is supported by DOD Breast Cancer Research Program Postdoctoral Fellowship W81XWH-12-1-0115. OJ is supported by NCI grant R21-CA177391.

## Additional information

### Competing interests

WCH: Consultant for Novartis. The other authors declare that no competing interests exist.

## Funding

| Funder | Grant reference number | Author |
|---|---|---|
| Congressionally Directed Medical Research Programs | W81XWH-12-1-0115 | Yaara Zwang |
| National Institutes of Health | U01 CA176058 | William C Hahn |
| National Cancer Institute | R21-CA177391 | Oliver Jonas |
| National Institutes of Health | R01 CA130988 | William C Hahn |

The funders had no role in study design, data collection and interpretation, or the decision to submit the work for publication.

## Author contributions

YZ, Conceptualization, Data curation, Formal analysis, Validation, Writing—original draft, Experimental design; OJ, Data curation, Formal analysis, Writing—original draft; CC, PB, Data curation, Formal analysis, Writing—review and editing; MLR, Conceptualization, Writing—review and editing; JGD, Conceptualization, Resources, Formal analysis, Writing—review and editing, Experimental design; FP, Resources, Experimental design; LT, H-TH, JW, YJH, Resources; JO, MD, MW, Data curation; MC, DER, RSL, Supervision; AL, Supervision, Writing—review and editing; NG, Supervision, Experimental design; WCH, Conceptualization, Supervision, Funding acquisition, Writing—original draft, Writing—review and editing

## Author ORCIDs

Yaara Zwang, http://orcid.org/0000-0002-5387-4663
Nathanael Gray, http://orcid.org/0000-0001-5354-7403
William C Hahn, http://orcid.org/0000-0003-2840-9791

## Ethics

Animal experimentation: Animals were maintained under conditions approved by the Institutional Animal Care and Use Committee at the Dana-Farber Cancer Institute (IACUC protocol #04-101) and at the Massachusetts Institute of Technology (IACUC protocol #0412-038-15).

# Additional files

## Supplementary files

• Supplementary file 1. Genes ranked as significant by either STARS or RIGER.

• Supplementary file 2. Details for shRNAs used for the five identified genes.

• Supplementary file 3. Structure, kinase selectivity and potency information of small molecule inhibitors targeting PIM2 or ZAK.

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
