## [Decision Letter]

Thank you for submitting your article "Synergistic interactions with PI3K inhibition that induce apoptosis" for consideration by *eLife*. Your article has been reviewed by three peer reviewers, one of whom, Chi Dang (Reviewer #1), is a member of our Board of Reviewing Editors and the evaluation has been overseen by Sean Morrison as the Senior Editor.

The reviewers have discussed the reviews with one another and the Reviewing Editor has drafted this decision to help you prepare a response to the critiques prior to a formal invitation for submission of a revised manuscript. We would like to determine whether it is realistic for you to adequately address the concerns raised by the reviewers.

Summary:

The manuscript from Zwang and colleagues reports a tour-de-force approach to identify synthetic lethal interactions with PI3K inhibition of PI3K mutant breast cancers using a novel approach of isolation dying/apoptotic cells transfected with an shRNA library. They describe the results of a screen in the MDA-MB-453 breast cancer cell line for shRNAs that induce cell death in combination with the PI3K inhibitor GDC0941. The authors then describe efforts to follow-up on the shRNAs for the 5 genes that were uncovered (PIM2, ZAK, TACC1, ZFR, ZNF565), using cDNA rescue approaches, pharmacological inhibitors of PIM2 and ZAK, and work with additional breast cancer cell lines. The authors conclude that the 5 genes encode proteins that represent potential targets and the work's significance is that it nominates potential combination therapy approaches that might add to PI3K inhibition in cancer treatment.

The studies and data are solidly pursued and presented, but the significance of the work for informing how the gene products are relevant to PI3K signaling in cancer are limited. The biological connections of the five genes and proteins to one another and to the PI3K signaling pathway seem uncertain, though some prior work suggests some possible relevance to PI3K signaling for PIM2, TACC1, and ZAK.

Essential revisions:

1) The in vivo work is weak with respect to the approaches used, the magnitude of the effects reported, and the specific assays and endpoints chosen. In the in vivo experiments, the authors need to provide more definitive information about the significance of the combination approach, relative to each mono-therapy approach, in inhibiting tumor progression and tumor size in the model used. Perhaps more significantly, It is not clear why the authors did not generate MDA-MB-452 and another relevant breast cancer cell line (e.g.,T47D, HCC1954, or HCC1937) that stably carry a Tet-inducible control shRNA or 2-3 different shRNAs for a few of the most promising identified genes and then assess whether shRNA induction +/- GDC0941 treatment has biologically and statistically significant tumor regression effects in xenografts.

2) Could the synthetic lethal interactions be generalized in other cancer types with oncogenic PI3K pathways? Will over expression of activated PIK3CA generate de novo dependency of the candidate genes (such as PIM2 or ZAK) in the cultured CAL-120 and MDA-MB-231 cell lines?

3) It seems that some evidence that the genes uncovered are part of a pathway or network could perhaps be uncovered through efforts to see whether ORF over-expression for PIM2, ZNF565, and TACC1 can rescue not only the effects of their own respective shRNAs, but can perhaps even rescue the effects of shRNAs for the other four genes found (e.g., does PIM2 over-expression not only rescue PIM2 shRNA effects, but also apoptosis seen with shRNAs for ZNF565, TACC1, ZAK, or ZFR?).

4) Do studies of the effects on gene expression in breast cancer cells following inhibition of the five genes by shRNA approaches or following inhibition of PIM2 and ZAK yield any evidence of enrichment of specific pathways or specific groups of genes?

5) The authors mention in the Discussion that the 5 genes are altered in 37% of breast cancers, but do not offer describe any specifics of whether the data suggest oncogenic activation (e.g., amplification and over-expression), tumor suppressor inactivation (e.g., nonsense, frameshift inactivating), or simply passenger mutations. How do the cancer data clarify how the genes are connected in any way to PI3K signaling or one another?

---

## [Author Response]

*Essential revisions:*

*1) The* in vivo *work is weak with respect to the approaches used, the magnitude of the effects reported, and the specific assays and endpoints chosen. In the* in vivo *experiments, the authors need to provide more definitive information about the significance of the combination approach, relative to each mono-therapy approach, in inhibiting tumor progression and tumor size in the model used. Perhaps more significantly, It is not clear why the authors did not generate MDA-MB-452 and another relevant breast cancer cell line (e.g.,T47D, HCC1954, or HCC1937) that stably carry a Tet-inducible control shRNA or 2-3 different shRNAs for a few of the most promising identified genes and then assess whether shRNA induction +/- GDC0941 treatment has biologically and statistically significant tumor regression effects in xenografts.*

In these studies, we used a microscale implantable device to test the effects of siRNA and small molecule inhibitors in vivo. This device has been shown by our collaborator to provide direct evidence of tumor responses in vivo (Jonas et al., 2015) and we have used the same system to interrogate the effects of small molecule inhibitors in patient-derived tumors (Hong et al., 2016). The major advantage of this device is that it allows more precise delivery of siRNA or small molecules, allows for better quantification of responses and facilitates the testing of combinations of treatments.

In this manuscript, we examined the effects of combining PI3K inhibition with suppression of the five identified genes in vivo. Figure 6 presents the reduction in the density of intact nuclei. Specifically, we measured a reduction of at least 23% in the density of intact nuclei when combining suppression of any of the five candidate genes with GDC0941 as compared to non-treated control mice (Figure 7). Using this data, we calculate the% effect from non-treated control siRNA (siGFP) for single treatments (each of the siRNAs or PI3K inhibition alone) and combined treatments (Figure 7). Based on the Bliss independence model, we also calculated the expected% effect for the combination based on the single treatments. For all siRNAs targeting any of the five candidate genes, the reduction in density of intact nuclei upon combination treatment was statistically significant (p <0.05) when compared to either single treatments. In addition, all combinations had a Bliss index significantly smaller than 1 indicating synergy

Author response image 1.Devices loaded with the indicated siRNAs were implanted into MDA-MB-453 xenografts of un-dosed mice or mice that were pre-dosed with GDC0941 (75mg/kg) for 5 days.Two days later, the devices were removed, xenograft tissue surrounding each reservoir was stained for Hematoxylin and Eosin., and density of intact nuclei was quantified.% reduction in intact nuclei upon PI3K inhibition is presented. Shown is the mean and SD of at least 4 replicates. B. Proportional effects of the indicated siRNAs with (red) or without (blue) PI3K inhibition as compared to the effect of siGFP in non-treated mice. Bars indicate SD from at least 4 replicates. The expected proportional effect for treatment combination (grey) was calculated according to the Bliss independence model using single treatment effects.**DOI:**
http://dx.doi.org/10.7554/eLife.24523.018

We include a description of the advantages of the microscale implantable device in Results, subsection “Suppression of the identified genes modifies the response to PI3K inhibition in vivo”. To clarify the strength and significance of our observations in vivo, we modified Figure 6: Figure 6 now shows the analysis shown here. The plot previously shown in Figure 6 is now shown in Figure 6—figure supplement 1. We also added a plot showing the% effect of combining PI3K inhibition with each of the siRNAs (Figure 6—figure supplement 1). We added the following text in subsection “Suppression of the identified genes modifies the response to PI3K inhibition in vivo”:

"Combining PI3K inhibition with any of the siRNAs targeting the five identified genes lead to a significant reduction in the density of intact nuclei as compared to non-treated control mice (Figure 6—figure supplement 1). […] All siRNAs targeting the five identified genes combined with PI3K inhibition exhibited greater effect then expected with a Bliss value significantly smaller then 1 (Figure 6)."

The reviewers also suggested that a Tet-inducible shRNA system could be used for such in vivo studies. We have extensive experience using such systems and have repeatedly found that it is necessary to clone cells to find clones that exhibit suitable inducible activity. Since we believe that one needs to examine many clones to ensure that the observed phenotype is correct, the number of experiments, which all include in vivo tumor experiments, would require nearly a year of work. Although we recognize that inducible shRNAs are attractive, our initial attempts to create such cell lines showed that polyclonal populations did not exhibit robust inducibility. We hope that our clarification of the microscale device that we used will allow the reviewers to agree that the statistically significant findings support the conclusions of this manuscript.

*2) Could the synthetic lethal interactions be generalized in other cancer types with oncogenic PI3K pathways? Will over expression of activated PIK3CA generate* de novo *dependency of the candidate genes (such as PIM2 or ZAK) in the cultured CAL-120 and MDA-MB-231 cell lines?*

To address this question, we examined two additional cancer types, ovarian cancer and glioblastoma multiforme (GBM). Of note, the main driver of PI3K pathway activation in GBM is PTEN inactivation, rather than PI3K activating mutations. We tested whether suppression of the five genes that we identified in the screen interacted with PI3K inhibition in one ovarian cancer cell line (Figure 3—figure supplement 1) and three GBM cell lines (Figure 3—figure supplement 1). All four cell lines undergo a proliferative arrest upon PI3K inhibition. Most of the interactions that we observed in breast cancer cell lines (Figure 3) were reproduced in EFO-21 (ovarian cancer; PIK3CA amplification). However, none of the GBM cell line exhibited similar response to combining suppression of any of the five candidate genes and PI3K inhibition. These experiments suggest that this effect is related to the mechanism of PI3K pathway activation as well as the particular lineage context.

We now include this new experimentation as Figure 3—figure supplement 1 and B and include the follow text in subsection “Confirming candidates in breast cancer cell lines.”:

"In addition to breast cancer cell lines, we also tested four additional non-breast cancer cell lines. […] However, the interactions we observed between PI3K inhibition and suppression of the five candidate genes in breast cancer, did not occur in the three tested GBM cell lines (all that harbor PTEN inactivation) (Figure 3—figure supplement 1)."

*3) It seems that some evidence that the genes uncovered are part of a pathway or network could perhaps be uncovered through efforts to see whether ORF over-expression for PIM2, ZNF565, and TACC1 can rescue not only the effects of their own respective shRNAs, but can perhaps even rescue the effects of shRNAs for the other four genes found (e.g., does PIM2 over-expression not only rescue PIM2 shRNA effects, but also apoptosis seen with shRNAs for ZNF565, TACC1, ZAK, or ZFR?).*

As was suggested, we tested whether PIM2, TACC1, and ZNF565 can interfere in the interaction between PI3K inhibition and the other respective 4 candidate genes suppression. We had shown in Figure 2 that these ORFs can rescue the effects of their respective shRNAs. We combined over expression of these ORFs with knock-down of all other four candidate genes and measured their ability to rescue cell from PI3K inhibition (Figure 2—figure supplement 1). Over-expression of ZNF565 did not rescue the suppression of any of the other 4 candidate genes, which suggests that ZNF565 cellular function is unique and cannot compensate for the suppression of the other four genes. Both PIM2 and TACC1 over-expression partially interfered in the interaction between PI3K inhibition and suppression of the other genes. Both PIM2 and TACC1 were shown before to contribute to cell survival upon PI3K signaling inhibition (Amaravadi, 2005; Cully et al., 2005). Hence the effect observed here is probably due to PIM2- or TACC1-mediated general increase in pro-survival signaling that alleviate the effect of PI3K inhibition, rather than specific interaction with the cellular function of other candidate genes.

We now include these experiments as Figure 2—figure supplement 1 as well as the following text in subsection “Validating candidates”:

"To investigate potential interactions between the five candidate genes, we utilized the rescue assay described here to test whether any of the genes can be rescued by another gene. […] suggesting that PIM2 or TACC1 generally contribute to cell survival in the presence of PI3K inhibition."

This result is also discussed in the Discussion section of the manuscript:

"In consonance with the known pro-survival function of PIM2 and TACC1 in the context of PI3K inhibition, we show here that their over-expression partially rescued the effect of combining PI3K inhibition with the other 4 candidate genes (Figure 2—figure supplement 1), further emphasizing the importance of combining their inhibition with PI3K inhibitors."

*4) Do studies of the effects on gene expression in breast cancer cells following inhibition of the five genes by shRNA approaches or following inhibition of PIM2 and ZAK yield any evidence of enrichment of specific pathways or specific groups of genes?*

We harnessed gene expression analysis to explore potential cellular mechanisms by which the five candidate genes synergize with PI3K inhibition, as well as potential interactions between them. To this end, we used five different control shRNAs and at least three different shRNAs to target each of the candidate genes. Seventy two hours after shRNA infection, cells were treated with either DMSO or GDC0941 (0.625uM) for 6 hours. Paired-end RNA-seq was conducted with at least 5 million reads per sample. Preprocessing and sequence alignment were done using GenePattern interface and included FASTQC, Trimmomatic, and TopHat. To avoid the off-target effects of individual shRNAs which affect multiple RNA molecules, we treated the different shRNAs that target the same candidate gene (or control shRNAs) as biological replicates. For each of the candidate genes, we compared gene expression upon it's suppression to control shRNAs using HTSeq followed by DESeq2 analysis. Plotted below, is the fold change (log2 transformed and centered) over control shRNAs of genes that exhibited significant differential expression (adjusted p value<0.05) upon suppression of the indicated candidate gene (Figure 8).

Author response image 2.RNA-seq data analyzed to identify differentially expressed genes upon suppression of candidate genes.(A,C) Centered log2 fold change over control shRNAs of genes that exhibit significant differential expression (Adjusted p value<0.05 according to DESeq2) upon suppression of the indicated candidate gene. B,D. Number of candidate genes whose suppression induces a significant differential expression of the genes plotted in A and C, respectively.**DOI:**
http://dx.doi.org/10.7554/eLife.24523.019

The majority of significantly differentially expressed genes were affected by the suppression of only one candidate gene in either DMSO or GDC0941 treatment (Figure 9). However, 4 genes were affected by the suppression of more than 2 candidate genes. MCM2, which is essential for DNA replication, was significantly down regulated upon suppression of PIM2, TACC1, ZAK and ZFR in the presence of GDC0941. Suppression of ZNF565 in the presence of GDC0941 also lead to MCM2 down-regulation, however, this was not statistically significant. The other three genes are FUS, SLC25A5, and TUBB. While each of them could contribute to the overall fitness of the cells, it is not clear if they specifically relate to any of the candidate genes that affected them, or to the PI3K pathway.

Author response image 3.Total number of genes that are significantly differentially expressed upon suppression of any of the five candidate genes under control or PI3K inhibition conditions.Number of genes that are affected by more than one candidate gene are detailed.**DOI:**
http://dx.doi.org/10.7554/eLife.24523.020

Next, we used DAVID (https://david.ncifcrf.gov/home.jsp) to query potential enrichment of functional annotations within the lists of significantly differentially expressed genes. We explored enrichment in molecular function, biological processes, and KEGG pathways (Figure 10). Most of the enriched annotations and pathways are too general, and do not suggest a specific mechanism by which the candidate genes interact with PI3K inhibition. One exception is the enrichment for RNA splicing and binding among the genes that are significantly differential expressed upon ZFR suppression. ZFR is a Zinc-finger RNA binding proteins, that was recently shown to be important for pancreatic cancer cells proliferation and invasion (Zhao et al., 2016). Therefore, it is possible that ZFR exerts it effect on cells partially through affecting RNA splicing.

Author response image 4.Functional annotation analysis of significantly differentially expressed genes.We listed the genes significantly affected by each of the candidate genes, under either control (**A**) or PI3K inhibition (**B**) condition, and queried these lists to identify enriched functions or pathways using DAVID. A Benjamini-Hochberg value of 0.05 was used as a threshold for enrichment.**DOI:**
http://dx.doi.org/10.7554/eLife.24523.021

Our conclusion is that this analysis did not reveal a clear evidence that indicates a common pathway or mechanism by which the candidate genes affect the fitness of the cells. However, since these experiments are not conclusive, we provide them here to answer the reviewer’s questions but have not incorporated them into the revised manuscript.

*5) The authors mention in the Discussion that the 5 genes are altered in 37% of breast cancers, but do not offer describe any specifics of whether the data suggest oncogenic activation (e.g., amplification and over-expression), tumor suppressor inactivation (e.g., nonsense, frameshift inactivating), or simply passenger mutations. How do the cancer data clarify how the genes are connected in any way to PI3K signaling or one another?*

In our analysis of the TCGA data, we included 1101 patient samples of invasive breast carcinoma. Of these, 54 samples (4.9%) that harbored deletion or RNA down-regulation of any of the five identified genes were excluded from the analysis. Of the remaining samples (1047), 37% had an amplification or RNA up-regulation of at least one of the five identified genes. Based on the available data, we failed to find significant co-occurrence or mutual exclusivity between any of the genes, suggesting that there is no clear interaction between these genes. When we classified the patients according to their PI3K pathway status of activity (determined according to the presence of either mutated or amplified PIK3CA, amplified or over-expressed HER2, or deleted or mutated PTEN), we also did not find a difference in the occurrence of alterations of the 5 genes between patients with or without PI3K pathway activation. It is important to note that data regarding the patients’ response to therapy and specifically to PI3K pathway inhibition, as well as clinical information regarding therapies preceding the collection of the sample are not available.

The following paragraph (Discussion section) was modified to include this information:

"Although there is no reason to expect that these 5 genes would be altered in human cancers, […] Moreover, there is no information regarding the response of the tumors analyzed by TCGA to PI3K pathway inhibition. As data from patients treated with PI3K inhibitors accumulates, it would be important to explore whether these genes are associated with response rate, or resistance."

References:

Hong, A.L., Tseng, Y.-Y., Cowley, G.S., Jonas, O., Cheah, J.H., Kynnap, B.D., Doshi, M.B., Oh, C., Meyer, S.C., Church, A.J., Gill, S., Bielski, C.M., Keskula, P., Imamovic, A., Howell, S., Kryukov, G.V., Clemons, P.A., Tsherniak, A., Vazquez, F., Crompton, B.D., Shamji, A.F., Rodriguez-Galindo, C., Janeway, K.A., Roberts, C.W.M., Stegmaier, K., van Hummelen, P., Cima, M.J., Langer, R.S., Garraway, L.A., Schreiber, S.L., Root, D.E., Hahn, W.C., Boehm, J.S., 2016. Integrated genetic and pharmacologic interrogation of rare cancers. Nat. Commun. 7. doi:10.1038/ncomms11987

Zhao, X., Chen, M., Tan, J., 2016. Knockdown of ZFR suppresses cell proliferation and invasion of human pancreatic cancer. Biol. Res. 49. doi:10.1186/s40659-016-0086-3